# Goal-directed vocal planning in a songbird

Anja T Zai[1,2]*, Anna E Stepien[1,2], Nicolas Giret[3], Richard HR Hahnloser[1,2]

[1]Neuroscience Center Zurich (ZNZ), University of Zurich and ETH Zurich, Zurich, Switzerland; [2]Institute of Neuroinformatics, University of Zurich and ETH Zurich, Zurich, Switzerland; [3]Institut des Neurosciences Paris-Saclay, UMR 9197 CNRS, Université Paris-Saclay, Saclay, France

## eLife assessment

This **important** work identifies a previously uncharacterized capacity for songbird to recover vocal targets even without sensory experience. The evidence supporting this claim is **convincing**, with technically difficult and innovative experiments exploring goal-directed vocal plasticity in deafened birds. This work has broad relevance to the fields of vocal and motor learning.

**Abstract** Songbirds' vocal mastery is impressive, but to what extent is it a result of practice? Can they, based on experienced mismatch with a known target, plan the necessary changes to recover the target in a practice-free manner without intermittently singing? In adult zebra finches, we drive the pitch of a song syllable away from its stable (baseline) variant acquired from a tutor, then we withdraw reinforcement and subsequently deprive them of singing experience by muting or deafening. In this deprived state, birds do not recover their baseline song. However, they revert their songs toward the target by about 1 standard deviation of their recent practice, provided the sensory feedback during the latter signaled a pitch mismatch with the target. Thus, targeted vocal plasticity does not require immediate sensory experience, showing that zebra finches are capable of goal-directed vocal planning.

*For correspondence:
zaia@ethz.ch

Competing interest: The authors declare that no competing interests exist.

## Introduction

Speech planning is an important part of human communication and the inability to plan speech is manifest in disorders such as apraxia. But to what extent is targeted vocal planning an entirely human ability? Many animals are capable of volitional control of vocalizations (**Brecht et al., 2019**; **Veit et al., 2021**), but are they also capable of planning to selectively adapt their vocalizations toward a target, such as when striving to reduce the pitch mismatch of a note in a song? Target-specific vocal planning is a cognitive ability that requires extracting or recalling a sensory target and forming or selecting the required motor actions to reach the target. Such planning can be covert or overt. Evidence for covert planning is manifest when a targeted motor change is executed without intermittent practice (**Costalunga et al., 2023**), for example, when we instantly imitate a word upon first hearing. Overt planning, by contrast, includes practice, but without access to the sensory experience from which target mismatch could be computed, for example, when we practice a piano piece by tapping on a table.

The vocal planning abilities in animals and their dependence on sensory experience remain poorly explored. Motor learning has been mostly studied in tasks where a skilled behavioral response must be produced on the spot, such as when a visual target must be hit by a saccade or by an arm reaching movement (**Brashers-Krug et al., 1996**; **Galea et al., 2011**; **Shmuelof et al., 2012**; **Krakauer et al., 2005**). In this context, motor planning has been shown to enhance motor flexibility as it allows

separation of motor memories when there are conflicting perturbations (*Sheahan et al., 2016*). However, for developmental behaviors such as speech or birdsong that rely on hearing a target early in life (*Konishi, 1985*; *Immelmann, 1969*), the roles of practice and of sensory feedback for flexible vocal control and for target-directed adaptation are unknown.

Recovery of a once-learned vocal skill could be instantaneous (covert) or it might require practice (overt). In support of the former, many motor memories are long-lasting (*Park et al., 2013*), for example, we can recall the happy-birthday song for years without practice. Some memories are even hard to get rid of such as accents in a foreign language. By contrast, practice-dependent, but feedback-independent recovery is argued for by arm reaching movements during use-dependent forgetting: following adaptation to biasing visual feedback, arm movements recover when the bias is either removed or the visual error is artificially clamped to zero (*Galea et al., 2011*; *Shmuelof et al., 2012*). One explanation put forward is that motor adaptation is volatile and has forgetting built-in (*Krakauer et al., 2005*; *Smith et al., 2006*), leading to practice-dependent reappearance of the original motor program even without informative feedback (*Smith et al., 2006*). Given these possibilities, we set out to probe songbirds' skills of recovering their developmental song target when deprived of either singing practice (to probe covert planning) or of sensory feedback (to probe overt planning).

Adult vocal performances in songbirds can be altered by applying external reinforcers such as WN stimuli (*Tumer and Brainard, 2007*; *Andalman and Fee, 2009*). When the reinforcer is withdrawn, birds recover their original song within hundreds of song attempts (*Tumer and Brainard, 2007*; *Canopoli et al., 2014*; *Warren et al., 2011*; *Hoffmann et al., 2016*). We argued that these attempts may be unnecessary and birds could recover their original performance by recalling either (1) the original motor program (*Aronov et al., 2008*; *Nottebohm et al., 1976*; *Prather et al., 2009*) or (2) its sensory representation (*Yazaki-Sugiyama and Mooney, 2004*; *Kojima and Doupe, 2007*; *Yanagihara and Yazaki-Sugiyama, 2016*) plus the mapping required for translating that into the original program (*Canopoli et al., 2014*; *Warren et al., 2011*; *Figure 1A*). These options might not need sensory feedback, which is argued for by birds' large perceptual song memory capacity (*Yu et al., 2020*). That is, birds' song practice may be mainly expression of deliberate playfulness (*Riters et al., 2019*), conferring the skill of vocal flexibility rather than serving to reach a target, evidenced by young birds that explore vocal spaces close to orthogonal to the song-learning direction (*Kollmorgen et al., 2020*) and that are already surprisingly capable of adult-like singing when appropriately stimulated (*Kojima and Doupe, 2011*).

## Results

To test whether birds can covertly recover a song syllable without practice, we first reinforced the pitch of a song syllable away from baseline and then we suppressed birds' singing capacity for a few days by muting their vocal output. We then unmuted birds and tested whether the song has covertly reverted back to the original target. We used syllable pitch as the targeted song feature because we found that birds did not reliably recover syllable duration in experiments in which we induced them to shorten or lengthen syllable duration (*Figure 1—figure supplement 1*).

We first drove pitch away from baseline by at least 1 standard deviation using a WN stimulus delivered whenever the pitch within a 16 ms time window locked to the targeted syllable was above or below a manually set threshold (*Figure 1B and C*, see 'Materials and methods'). We muted these WNm (WN reinforced and muted) birds by implanting a bypass cannula into the abdominal air sac (see 'Materials and methods'). While muted, air is leaking from the abdominal air sac and as a result, sub-syringeal air pressure does not build up to exceed the threshold level required for the self-sustained syringeal oscillations (*Elemans et al., 2015*) that underlie singing. Physical absence of such oscillations essentially strips muted birds from all pitch experience. In some cases, the bypass cannula got clogged during the muted period and birds were spontaneously unmuted, allowing them to produce a few songs before we reopened the cannula (*Figure 1C–G*).

We quantified recovery in terms of normalized residual pitch (NRP) to discount for differences in the amount of initial pitch shift where NRP = 0% corresponds to complete recovery and NRP = 100% corresponds pitch values before withdrawal of reinforcement ($R$) and thus no recovery. After spending 5.1 ± 1.6 days (range 3–8 days, N = 8) in the muted state and upon unmuting, WNm birds displayed an average NRP of 89%, which was far from baseline (p=6.2 · 10$^{-8}$, tstat = -23.6, N = 8 birds, two-sided *t*-test of H0: NRP = 0%, songs analyzed in 2 hr time window – early ($E$), see 'Materials and methods',

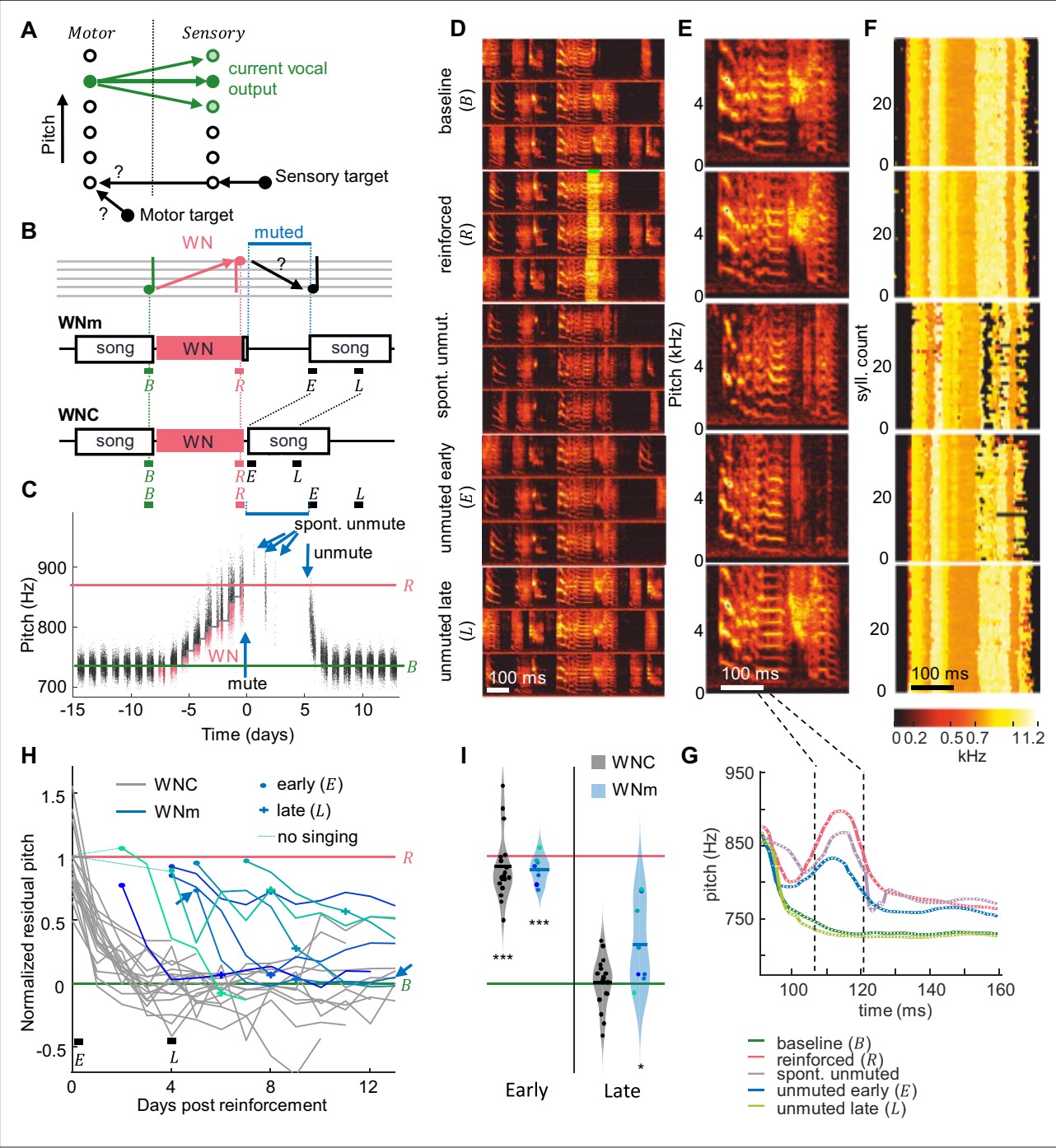

**Figure 1.** Recovery of pitch target requires practice. (**A**) Two hypotheses on birds' ability to recover a song target away from their current vocal output (green circles, motor states on the left, sensory states on the right, shading represents probabilities): Either they could recall the motor target and reactivate it without practice or they could recall a sensory target plus the neural mapping (black arrows) required to transform it into a motor state. (**B**) WNm birds were first pitch-reinforced using white noise (WN), then muted, and subsequently unmuted. WN was delivered when the pitch of the target syllable was either below (as exemplified here) or above a threshold. Pitch recovery from the reinforced (*R*) state toward the baseline (*B*) target is evaluated in early (*E* no practice) and late (*L*, with practice) analysis windows (all windows are time-aligned to the first 2 hr of songs after withdrawal of reinforcement, *E*) and compared to recovery in unmuted control birds (WNC). (**C**) Syllable pitches (dots, red = reinforced syllables) of an example bird that while muted recovered only about 27% of pitch difference to baseline despite three spontaneous unmuting events (arrows). (**D**) Same bird, spectrograms of example song motifs from five epochs: during baseline (*B*), reinforcement (*R*) with WN (green bar), spontaneous unmuting (spont. unmut.), and during permanent unmuting (early – *E* and late – *L*). (**E**) Example syllables from same five epochs. (**F**) Stack plot of pitch traces (pitch indicated by color, see color scale) of the first 40 targeted syllables in each epoch ('reinforced': only traces without WN are shown). (**G**) Average pitch

*Figure 1 continued on next page*

*Figure 1 continued*

traces from (F), revealing a pitch increase during the pitch-measurement window (dashed black lines) and pitch recovery late after unmuting. (**H**) WNm birds (blue lines, N = 8) showed a normalized residual pitch (NRP) far from zero several days after reinforcement (circles indicate unmuting events, arrow shows bird from **C**) unlike WNC birds (gray lines, N = 18). Thin dashed lines indicate the two initial birds that were not given reinforcement-free singing experience before muting (see 'Materials and methods'). (**I**) Violin plots of same data restricted to early and late analysis windows (***p<0.001, *p<0.05, two-tailed *t*-test of NRP = 0).

The online version of this article includes the following figure supplement(s) for figure 1:

**Figure supplement 1.** Birds rapidly recover pitch but not duration after reinforcement learning.

*Figure 1*), suggesting that in the muted state, birds are unable to recover their pre-reinforced songs. The average NRP in WNm birds was comparable to that of unmanipulated control (WNC) birds within the first 2 hr after withdrawal of the reinforcer (average NRP = 91%, p=3.7 · 10⁻¹¹, tstat = 14.8, two-sided *t*-test for NRP = 0%, N = 18 WNC birds). Indeed, during 5 days without song practice, birds recovered no more pitch distance than birds normally do within the first 2 hr of release from reinforcement (p=0.82, tstat = -0.23, N = 8 WNm and N = 18 WNC birds, two-sided *t*-test). In WNm birds, there was no correlation between the NRP in the early window and the time since the muting surgery (correlation coefficient R=0.26, p=0.53), suggesting that the lack of pitch recovery while muted was not due to a lingering burden of the muting surgery. These findings did not sensitively depend on the size of the analysis window — we also tested windows of 4 and of 24 hr.

Subsequently, after 4 days of unmuted singing experience (roughly 9 days after withdrawal of WN), WNm birds displayed an average NRP of 30%, which was significantly different from the average NRP within the first 2 hr after unmuting (p=3 · 10⁻⁴, tstat = 4.83, N = 8 birds, two-tailed *t*-test early (*E*) vs. late (*L*) time window) but still significantly different from zero (p=0.04, tstat = 2.59, N = 8 birds, two-tailed *t*-test, late (*L*) time window). The amount of recovery was neither correlated with the number of renditions sung between early and late windows (R=0.03, p=0.95), nor with the duration the birds were muted (R=–0.50, p=0.20), nor with the time since they last sung the target song before reinforcement (R=–0.43, p=0.29), suggesting the limiting factor for recovery was neither the amount of song practice nor the recovery time from the muting surgery (although for the latter there was a trend). Overall, these findings rule out covert planning in muted birds and suggest that motor practice is necessary for recovery of baseline song.

Next, we tested whether motor experience but not sensory experience is necessary for overt recovery, similar to arm reaching movements that can be restored without guiding feedback (*Galea et al., 2011*; *Galea et al., 2015*). In a second group of birds, we provided slightly more singing experience (*Figure 2*). Instead of muting, WNd birds were deafened through bilateral cochlea removal immediately after the end of WN reinforcement. This latter manipulation does not suppress the act

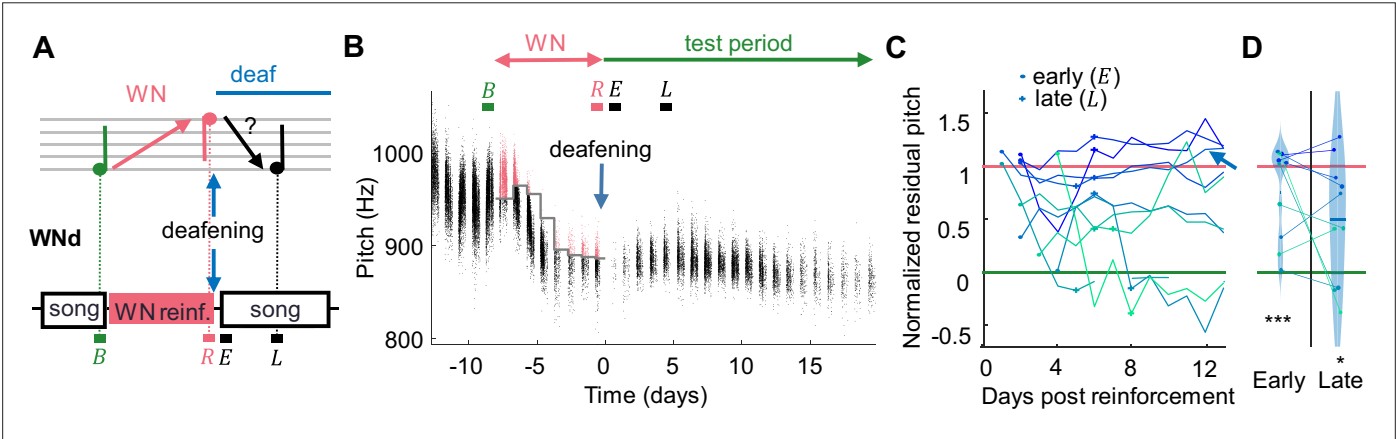

**Figure 2.** Recovery of pitch target is impaired after deafening. (**A**) WNd birds were first pitch-reinforced using white noise (WN) and then deafened by bilateral cochlea removal. Analysis windows (letters) as in *Figure 1*. (**B**) Syllable pitches (dots, red = reinforced syllables) of example WNd bird that shifted pitch down by *d'* = -2.7 during WN reinforcement and subsequently did not recover baseline pitch during the test period. (**C**) WNd birds (N = 10) do not recover baseline pitch without auditory feedback (circles = early window after deafening events, cross = late). (**D**) Violin plots of same data restricted to early and late analysis windows, lines connect individual birds (***p<0.001, *p<0.05, two-tailed *t*-test of NRP = 0).

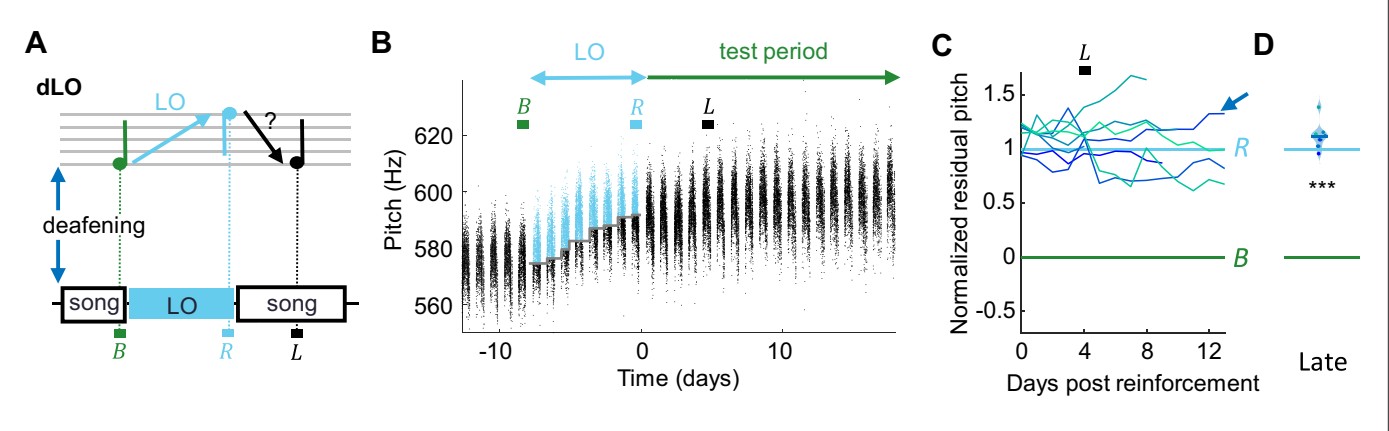

**Figure 3.** Deaf birds do not recover pitch target after light-induced mismatch. (**A**) dLO birds were first deafened and then pitch-reinforced using a brief light-off (LO) stimulus. Analysis windows (letters) as in *Figure 1*. (**B**) Syllable pitches (dots, blue = LO-reinforced syllables) of example dLO bird that shifted pitch up by *d'* = 3.5 within a week, but showed no signs of pitch recovery during the test period. (**C**) dLO birds (N = 8) do not recover baseline pitch without auditory feedback. (**D**) Violin plots of same data restricted to the late analysis window (***p<0.001, two-tailed *t*-test of NRP = 0).

of singing as does muting, but it eliminates auditory feedback from singing. Deaf birds could gain access to some pitch information via somatosensory stretch and vibration receptors and/or air pressure sensing (*Suthers et al., 2002*). Our aim was to test whether such putative pitch correlates are sufficient for recovery of baseline pitch (*Figure 2A*). However, in the deaf state, WNd birds did not recover baseline pitch even after 4 days of song practice: on the fifth day (late, *L*) after deafening, their average NRP was still 50%, which was different from zero (p=0.03, tstat = 2.73, two-tailed *t*-test of H0: NRP = 0%, N = 10, *Figure 2D*) and significantly larger than the average NRP of WNC birds on the fifth day since withdrawal of reinforcement (difference in NRP = 49%, p=0.003, tstat = 3.34, df = 26, N = 10 WNd and N = 18 WNC birds, two-tailed *t*-test).

We speculated that the lack of pitch recovery in WNd birds could be attributable to the sudden deafening experience, which might be too overwhelming to uphold the plan to recover the original pitch target. WN deaf birds did not sing for an average of 2.3 ± 1.1 days (range 1–4 days) after the deafening surgery, which is a strong indication of an acute stressor (*Yamahachi et al., 2020*). We thus inspected a third group of birds (dLO, *Figure 3*) taken from *Zai et al., 2020a* that learned to shift pitch while deaf and that underwent no invasive treatment between the pitch reinforcing experience and the test period of song recovery.

dLO birds were first deafened, and after they produced stable baseline song for several days, their target syllable pitch was reinforced using pitch-contingent light-off (LO) stimuli, during which the light in the sound recording chamber was briefly turned off upon high- or low-pitch syllable renditions (*Zai et al., 2020b*). dLO birds displayed an average NRP of 112% on the fifth day since release from LO, which was significantly different from zero (p=3.7 · $10^{-8}$, tstat = 25.4, N = 8 birds, two-tailed *t*-test of H0: NRP = 0) and was larger than the NRP in WNC birds on the fifth day since release (p=1.3 · $10^{-13}$, tstat = 14.9, df = 24, N = 8 dLO and N = 18 WNC birds, two-sided *t*-test). Thus, dLO birds were unable to recover baseline pitch, suggesting that song recovery requires undiminished sensory experience, which includes auditory feedback.

Deaf birds' decreased singing rate could not explain their lack of pitch recovery. Deaf birds sang less during the first 2 hr since release of reinforcement (early) than control birds: 87 ± 59 motif renditions for WNd and 410 ± 330 renditions for dLO compared to 616 ± 272 renditions for WNC birds. Also, WNd birds sang only 4,300 ± 2,300 motif renditions between the early and late period compared to the average of 11,000 ± 3,400 renditions that hearing WNC birds produced in the same time period. However, despite these differences, when we inspected WNd birds' behavior 9 days after the early window, when they sung on average 12,000 ± 6,000 renditions, their NRP was still significantly different from zero (NRP = 0.37, p = 0.007, tstat = 3.47, df = 9). Thus, even after producing more practice songs than control birds, deaf birds did not recover baseline pitch and so the number of songs alone cannot explain why deaf birds do not fully recover pitch. We conclude that auditory experience seems to be necessary to recover song.

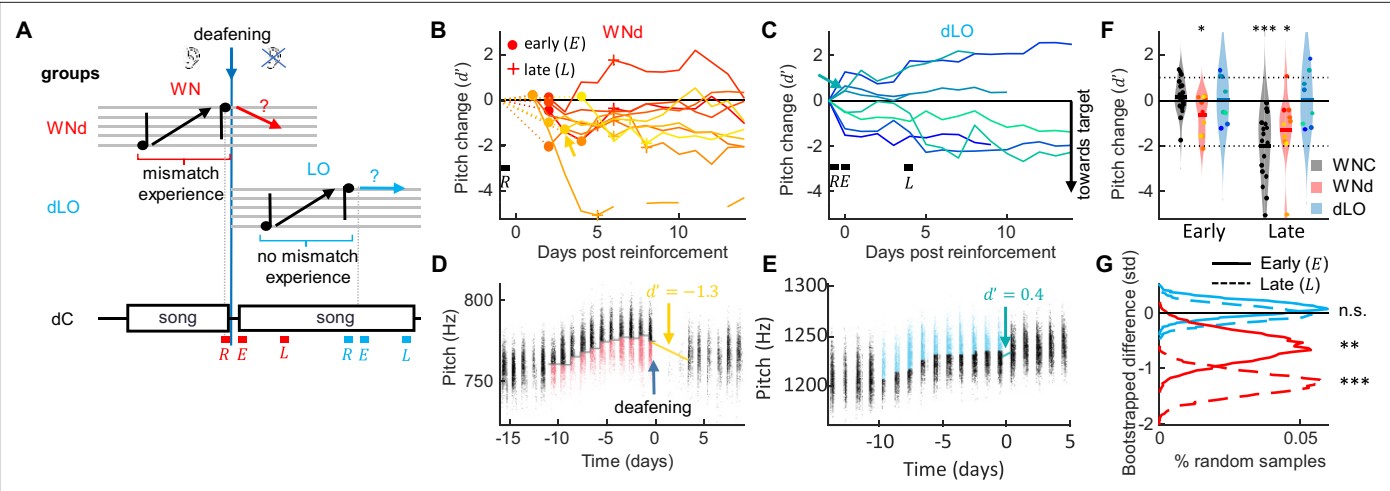

**Figure 4.** Target mismatch experience is necessary for revertive pitch changes. (**A**) WNd birds heard a target mismatch during reinforcement whereas dLO birds did not. dC birds were not pitch reinforced, their analysis windows (letters as in *Figure 1*) matched those of manipulated birds in terms of time since deafening. (**B, C**) Pitch change between the last 2 hr of reinforcement ($R$) and the pitch of successive days aligned to the first 2 hr of song after withdrawal of reinforcement in std for WNd (red, **B**) and dLO (blue, **C**) birds. Early and late windows are marked with markers (dots, crosses) in (**B**) and letters ($E$, $L$) in (**C**). Curves are plotted such that pitch changes toward the target are pointing down (see 'Materials and methods'). Example birds shown in (**D, E**) are marked with arrows. (**D, E**) Syllable pitches (dots, red = WN reinforced, blue = LO-reinforced syllables) of example WNd (**D**) and dLO (**E**) bird. (**F**) WNd (red) perform both early and late pitch changes in the direction of the baseline target (by about one standard deviation, * p<0.05, ***p<0.001, one-tailed *t*-test, N=10 WNd birds and N = 8 dLO birds), similar to WNC (gray) and unlike dLO (blue) birds without mismatch experience. Dotted lines mark y-range displayed in (**G**) Bootstrapped pitch differences between reinforced WNd (red) and dLO (blue) and 10,000 times randomly matched dC birds, shown for early (solid line) and late (dashed line) analysis windows. The stars indicate the bootstrapped probability of a zero average pitch difference between reinforced and dC birds (n.s. not significant, **p<0.01, ***p<0.001, N=10 WNd birds and N = 8 dLO birds).

That song practice and sensory experience are required for full recovery of song does not imply that without experience, birds are incapable of making any targeted changes to their songs at all. We therefore inspected birds' fine-grained vocal output and whether they changed their song in the direction of baseline when deprived of sensory experience. We hypothesized (*Figure 4A*) that when birds experience a target mismatch during reinforcement (i.e., they hear that their song deviates from the target), they plan to recover the pitch target, and a portion of this plan they can execute without feedback. If, by contrast, they have no mismatch experience before deafening, they will make no corresponding plan. Hence, we predicted that WNd birds that experienced a pitch mismatch during reinforcement and before deafening would slightly revert their song toward baseline even in the absence of auditory feedback. By contrast, dLO birds that did not experience a mismatch because they did not hear their song while it was reinforced would not revert toward the target (*Figure 4A*).

Indeed, WNd birds changed their pitch significantly toward baseline already in the first 2 hr of their singing since release from reinforcement (relative to the pitch from the last 2 hr during reinforcement). We quantified local pitch changes in terms of the *d'* sensitivity of signal detection theory (which is independent of shift magnitude) and found *d'* = -0.60 (p = 0.03, tstat = -2.19, df = 9, N = 10 WNd birds, one-sided *t*-test of H0: *d'* = 0). A significant reversion toward pitch baseline was still evident after 4 days of practice (*d'* = -1.27, p = 0.02, tstat = -2.35, N = 10 WNd birds, one-sided *t*-test, *Figure 4B, D and F*), showing that pitch reversion in deaf birds is persistent. Because the average pitch shift in WNd birds was on the order of one standard deviation (*d'* ≃ 1), we conclude that without auditory experience, birds are able to perform target-directed pitch shifts of about the same magnitude as their current exploratory range (i.e., the denominator of the *d'* measure).

In contrast, dLO birds showed no signs of reverting pitch, neither in the first 2 hr since release of reinforcement (*d'* = -0.13, p = 0.36, tstat = -0.37, df = 7, N = 8 birds, one-sided *t*-test), nor after 4 days of practice (*d'* = -0.08, p = 0.43, tstat = -0.18, df = 7, N = 8 birds, one-tailed *t*-test, *Figure 4C, E and F*).

The singing rate does not explain why deaf birds with mismatch experience partially revert their song toward baseline, unlike deaf birds without mismatch experience. WNd birds sang less during the first 2 hr after reinforcement (early) than both control birds (p=2.3 · 10⁻⁶, tstat = 6.02, df = 26, N

= 10 WNd and N = 18 WNC birds, two-sided *t*-test) and dLO birds (p=0.008, tstat = 3.06, df = 16, N = 8 dLO birds, two-sided *t*-test), unlike dLO birds that sang similar amounts as WNC birds (p = 0.11, tstat = 1.67, df = 24, two-sided *t*-test). If the number of songs were to determine the rate of recovery, we would have seen the opposite effect (dLO birds should recover similar amounts as WNC birds and significantly more than WNd birds). In conclusion, singing rate does not explain the difference between WNd and dLO birds.

To discount for the effect of time elapsed since deafening and quantify the change in pitch specifically due to reinforcement, we bootstrapped the difference in *d'* between dLO/WNd birds and a new group of dC birds that were deafened but experienced no prior reinforcement (see 'Materials and methods'). To discount for possible influences of circadian pitch trends, we assessed early and late pitch changes in reinforced birds and in dC birds in 2 hr time windows separated by multiples of 24 hr (and again flipped pitch changes in birds that were reinforced to decrease pitch, see 'Materials and methods'). In agreement with the findings above, we found that significant reversion toward baseline was only seen in WNd birds and very consistently so (*Figure 4G*, *Supplementary file 1*), showing that prior experience of a target mismatch is necessary for pitch reversion independent of auditory feedback.

We further validated our finding using a linear mixed effect model on the combined NRP data of all groups (see 'Materials and methods'), which confirmed our previous findings: we did not find a significant effect of the time without practice between $R$ and $E$ windows on the NRP in the $E$ window (fixed effect –0.04, p = 0.2), confirming that birds do not recover without practice. Neither deafening nor muting had a significant effect by itself but the interaction between deafening and time (late) was associated with an NRP increase of 0.67 (fixed effect, p = $2 \cdot 10^{-6}$), demonstrating that deaf birds are significantly further away from baseline (NRP = 0) than hearing birds in late windows, thereby confirming that birds require auditory feedback to recover a distant pitch target. Importantly, we found that mismatch experience was associated with a significant fixed effect of –0.37 on the NRP (fixed effect toward the target, p = 0.006), supporting our finding that limited vocal plasticity is possible even in the absence of auditory feedback.

Our results thus argue for a model of song maintenance in which birds extract from target mismatch experience a plan of reducing the mismatch. Without practice and auditory experience, birds cannot reach a distant motor target (*Figure 5A*). With practice and without auditory experience, they can make small changes toward a target, which we refer to as the planning range. Auditory experience allows them to consolidate the small changes such that step-by-step they can reach even a distant target (*Figure 5B*).

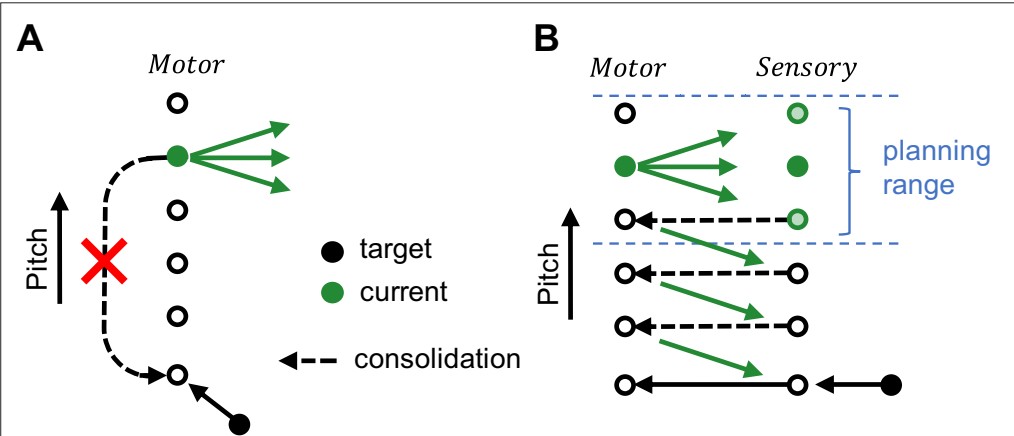

**Figure 5.** Schematic illustrating the goal-directed planning of vocal changes. (**A**) Without practice, birds cannot recover a distant motor target (black filled circle) far away from the current motor output (green filled circle). (**B**) Without auditory experience, birds can make motor changes (green arrows) toward a target within a small range, we refer to this range as the (overt) planning range (blue). To recover a distant target (black filled circle) beyond the planning range, birds need auditory experience (green circles under Sensory), presumably to consolidate (dashed arrows) the overt motor changes.

# Discussion

Our work shows that recent auditory experience can drive motor plasticity even while an individual is deprived of such experience, that is, zebra finches are capable of overt vocal planning. But to reach a distant vocal target beyond the pitch range they have recently produced necessitates auditory feedback, which sets a limit to zebra finches' overt planning ability.

Our insights were gained in deaf birds, and we cannot rule out that deaf birds could gain access to pitch information via somatosensory-proprioceptive sensory modalities. However, such information, even if available, cannot explain the difference between the 'mismatch experience' (WNd) and the 'no mismatch experience' (dLO) groups, which strengthens our claim that the pitch reversion we observe is a planned change and not merely a rigid motor response (as in simple use-dependent forgetting; *Galea et al., 2011*; *Shmuelof et al., 2012*). Also, it is unlikely that dLO birds' inability to recover baseline pitch is somehow due to our use of a reinforcer of a non-auditory (visual) modality since somatosensory stimuli do not prevent reliable target pitch recovery in hearing birds (*Sober and Brainard, 2009*). Thus, the overt planning ability is an active experience-dependent process.

In our two-stage model, recovery of a developmentally learned vocal target is controlled by two nested processes, a highly flexible process with limited scope ($d' \simeq 1$, *Figures 4 and 5*), and a dependent process enabled by experience of the former. Such motor learning based on separate processes for acquisition and retention is usually referred to as motor consolidation (*Brashers-Krug et al., 1996*; *Fenn et al., 2003*; *Karni and Sagi, 1993*). Accordingly, the flexible process of acquisition or planning as we find is independent of immediate sensory experience, but the dependent process (consolidation of the flexible process) requires experience. Perhaps then, it is the sensory experience itself that is consolidated, and therefore, consolidation of sensory experience may be a prerequisite for extensive planning.

We cannot distinguish the overt planning we find from a more complex use-and-experience-dependent forgetting since we only probed for recovery of pitch and did not attempt to push birds into planning pitch shifts further away from baseline. Evidence for more flexible planning is provided by the pitch matching skills of nightingales (*Costalunga et al., 2023*). Interestingly, although nightingales can reach without practice even distant pitch targets, the targets in *Costalunga et al., 2023* were also located within the extent of nightingale's recent song practice, so also satisfied $d' \simeq 1$. Perhaps then, our two-stage model of song plasticity of planning and consolidation in *Figure 5* applies more broadly in songbirds and not just in zebra finches.

Consolidation in motor learning generally emerges from anatomically separated substrates for learning and retention (*Galea et al., 2011*). Such separation also applies to songbirds. Both reinforcement learning of pitch and recovery of the original pitch baseline depend on the anterior forebrain pathway and its output, the lateral magnocellular nucleus of the anterior nidopallium (LMAN) (*Warren et al., 2011*). LMAN generates a pitch bias that lets birds escape negative pitch reinforcers and recover baseline pitch when reinforcement is withdrawn (*Andalman and Fee, 2009*), thus is likely involved in planning. This pitch bias is consolidated outside of LMAN (*Warren et al., 2011*; *Tian and Brainard, 2017*) in a nonlinear process that is triggered when the bias exceeds a certain magnitude (*Tachibana et al., 2022*). This threshold magnitude is roughly identical to the planning limit we find ($d' \simeq 1$), suggesting that birds' planning limit arises from the consolidation of LMAN-mediated motor plasticity. Although it remains to be seen whether LMAN is capable of executing motor plans without sensory feedback, our work provides a new perspective on the neural basis of birdsong learning and consolidation in and around LMAN.

The formation of a planned motor change may not require LMAN itself because pharmacological suppression of LMAN sets the bias to zero, but upon removal of output suppression, the pitch of the song syllable that was targeted by reinforcement jumps by about 1% away from the reinforced pitch zone (*Charlesworth et al., 2012*), which corresponds to about $d' = 1$, about the planning limit we find. Originally, this jump was interpreted as evidence of functional connectivity or an efference copy between the anterior forebrain pathway of which LMAN is part of and some other unspecified variability-generating motor area. However, in our view, a simpler explanation requiring neither functional connectivity nor efference copy is that LMAN is involved in putting a plan into action, which in that case is to produce syllable variants that are unaffected by WN.

Zebra finches' ability to plan directed song changes could hinge on song memories that feed into LMAN and that could drive neurons there to produce diverse perceptual song variants. LMAN neurons

are selective for the bird's own song but not the target song (*Yazaki-Sugiyama and Mooney, 2004*; *Kojima and Doupe, 2007*), which makes them well suited for executing song plans within the range of recent experience (i.e., if the song is outside recent experience, it elicits no LMAN response and so does not gain access to planning circuits). Furthermore, LMAN neurons show mirrored activity, that is, similar activity when a zebra finch produces a vocal gesture and when it hears the same gesture played through a loudspeaker (*Hanuschkin et al., 2013*; *Oztop et al., 2013*). This mirrored activity has been argued to be involved in translating an auditory target into the corresponding motor command, also known as an inverse model (*Giret et al., 2014*). Mirroring in LMAN was observed across the song variability generated over a period of several hours, which is about the same as the experience-dependent pitch planning limit we find. Zebra finches could thus transform a desired pitch change into the corresponding motor plan via LMAN's aligned sensory and motor representations of recent vocal output.

In a broader context of motor recovery, birds' failure to recover baseline pitch without guiding sensory feedback agrees with reports that binary reinforcement (as we used) slows down or prevents forgetting of the adapted behavior (*Shmuelof et al., 2012*). However, whereas forgetting is fast when sensory errors affect arm movements (*Shmuelof et al., 2012*), the contrary applies to birdsong, where pitch learning from artificial sensory errors is slower and less forgotten (*Sober and Brainard, 2009*) than is pitch learning from binary reinforcement (*Tumer and Brainard, 2007*; *Canopoli et al., 2014*). Hence, the commonality of short-term visuo-motor adaptation and of birdsong maintenance is that slow learning leads to slow forgetting, regardless of whether it is due to sensory errors or reinforcement. Such conclusion also agrees with observations that zebra finch song does not recover to pre-manipulated forms, both after restoring auditory feedback after long-term (>5 months) deprivation (*Zevin et al., 2004*) and after restoring normal syrinx function after long-term (16 weeks) manipulation with beads (*Hough and Volman, 2002*), suggesting that song can spontaneously recover only within some limited time since it was manipulated.

Our observations in zebra finches could be relevant to other species including humans. The planning abilities we find bear resemblance to human motor imagery for movement learning, which is most effective when subjects already show some competence for the movements to be learned (*Mulder et al., 2004*), suggesting a recall-dependent process. Naively, human vocal flexibility seems superior to that of zebra finches since we can flexibly change sound features such as loudness, pitch, and duration to convey emotional state or to comply with the tonal and rhythmical requirements of a musical piece (*Dichter et al., 2018*; *Belyk and Brown, 2016*), whereas zebra finches produce more subtle modulations of their songs, for example, when directing them to a female (*Stepanek and Doupe, 2010*). Nevertheless, a limit of human vocal flexibility is revealed by non-native accents in foreign languages, which are nearly impossible to get rid of in adulthood. Thus, a seeming analogous task to feedback-dependent learning of zebra finch song, in humans, is to modify developmentally learned speech patterns.

Our findings help elucidate the meaning of song signals in songbirds and the evolutionary pressures of singing. Because zebra finches seem incapable of large jumps in performance without practice, their current song variants are indicative of the recent song history, implying that song is an honest signal that zebra finches cannot adapt at will to deceive a receiver of this signal. Hence, if high pitch has either an attractive or repelling effect on another bird, a singer must commit to being attractive or repulsive for some time. In extension, we speculate that limited vocal flexibility increases the level of commitment to a group and thereby strengthens social cohesion.

## Materials and methods

All experimental procedures were in accordance with the Veterinary Office of the Canton of Zurich (licenses 123/2010 and 207/2013) or by the French Ministry of Research and the ethical committee Paris-Sud and Centre (CEEA no. 59, project 2017-12).

### Subjects

We used in total 76 birds. All birds were 90–300 days old (except one 853-day-old control bird) and were raised in the animal facility of the University of Zurich or in Saclay. During recording, birds were housed in single cages in custom-made sound-proof recording chambers equipped with a wall

microphone (Audio-Technica Pro4 and 2), a loudspeaker. The day/night cycle was set to 14/10 hr except for one muted bird that was in constant light due to a technical problem.

## Song recordings

Vocalizations were saved using custom song-recording software (Labview, National Instruments Inc). Sounds were recorded with a wall microphone and digitized at 32 kHz. In all birds, we recorded baseline vocal activity for at least 3 days before doing any manipulation (deafening or pitch reinforcement).

## Pitch reinforcement

We calculated pitch (fundamental frequency) as described in *Canopoli et al., 2014*. To provide pitch reinforcement in real time, we used a two-layer neural network trained to detect a manually clustered syllable containing a harmonic stack (*Yamahachi et al., 2020*). We evaluated the fundamental frequency of that syllable in a 16–24 ms time window following detection. For pitch reinforcement, we either broadcast a 50–60-ms-long WN stimulus through a loudspeaker or briefly switched off the light in the isolation chamber for 100–500 ms (LO) when pitch was below or above a manually set threshold. The WN/LO stimulus onset occurred 7 ms after the pitch calculation offset. We performed cumulative pitch shifts across several days by adjusting the pitch threshold for WN/LO delivery each day, usually setting it close to the median value of the previous day. Sometimes the threshold was set more than once during a day, in this case we set it close to the median of the pitch values measured so far during that day. All birds were shifted by at least 1 standard deviation ($d' > 1$, see 'Pitch analysis').

Reported pitch values were collected as above, except in muted birds that directly after unmuting produced syllables of lower amplitude and with distorted spectral features (e.g., *Figure 1C, E and F*), which resulted in frequent mis-detections by the neural network. In muted birds, we therefore performed semi-automatic (manually corrected) syllable detection and computed pitch at a fixed time lag after syllable onset. Despite deafening leading to degradation of birds' song (*Lombardino and Nottebohm, 2000*), syllable detection and pitch calculation were still possible in all deaf birds (birds were recorded during 13–50 days after deafening surgery, age range 90–300 dph, N = 44 birds). Since pitch shifting was balanced in all deaf bird groups (the same number of birds were up- and down-shifted), systematic changes in pitch post deafening (*Lombardino and Nottebohm, 2000*) will average out and so would not affect our findings.

## Duration reinforcement

Duration reinforcement was performed similarly as pitch reinforcement, but instead of measuring the pitch of a targeted syllable, we measured the duration of a targeted song element (either a syllable, a syllable plus the subsequent gap, or just a gap). Onsets and offsets of the targeted element were determined by thresholding of the root-mean-square (RMS) sound amplitude.

## Bird groups

### WN control (WNC)

Eighteen birds in the control group underwent WN pitch reinforcement (10/18 up-shifted, 8/18 down-shifted). Thereafter, the WN stimulus was withdrawn, and no further experimental manipulation took place.

### WN muted (WNm)

In eight birds, we first reinforced pitch using WN auditory stimuli and then we reversibly muted the birds by performing an airsac cannulation.

Normally, when WN stimuli are contingent on low-pitch renditions, birds tend to shift the pitch up, and in 5/6 birds this was indeed the case. However, one bird shifted the pitch down, in an apparent appetitive response to WN, this bird responded appetitively also when the WN contingency was changed, resulting in a net upward shift at the end of the WN period (see also *Yamahachi et al., 2020*). In two birds, we targeted high-pitch variants and these birds shifted the pitch down, as expected. Thus, in total, in 6/8 birds (including the bird with the apparent appetitive response), we drove the pitch up, and in 2/8 birds, we drove the pitch down.

Two birds underwent the muting surgery directly after withdrawal of WN stimuli. We observed that the two birds had recovered a mere 10% and –6% of their total WN-driven pitch change (*Figure 1H*).

We hypothesized that unreinforced singing would initiate the song recovery process in WNm birds that we assumed birds might be able to accomplish while mute. Therefore, we allowed the subsequent 6/8 WNm birds (four up-shifted and two down-shifted) to sing a few hundreds of target syllables without reinforcement prior to muting them. During on average 4 hr 51 min (range 10 min to 14 hr), these latter birds produced on average 649 song motifs (56, 100, 400, 458, 480, and 2400 motifs) without WN; the example bird shown in *Figure 1C* produced 56 song motifs within 11 min during the 30 min it was allowed to sing without aversive reinforcement.

### WN deaf (WNd)

Ten birds were first pitch reinforced (5/10 were up-shifted and 5/10 down-shifted) with WN, and then they were deafened immediately after by bilateral cochlea removal. WNd birds started to sing on average 3 ± 1 days after deafening (range 2–5 days) and were recorded for at least 15 days after the deafening surgery.

### Deaf LO (dLO)

Here, 8/10 birds from *Zai et al., 2020b* were recorded after the reinforcement period and we analyzed the associated data. These birds were first deafened by bilateral cochlea removal, then they underwent pitch reinforcement with light-off (LO) stimuli that acts as an appetitive stimulus in deaf birds. The lamp in the recording chamber was switched off for 100–500 ms when the pitch was either above or below a manually set threshold (daily threshold adjustment followed the same procedure as for WNm birds). 3/8 birds received LO for low-pitched syllables and 5/8 birds for high-pitched syllables. One of the birds that received LO for high-pitched syllables changed its pitch away from LO instead of toward it, thus we ended up with a balanced data set with 4/8 birds shifting pitch up and 4/8 birds shifting down. dLO birds were recorded for at least 5 days after the deafening surgery. Details of light-induced pitch shifting are described in *Zai et al., 2020b*.

### Deaf control (dC)

We analyzed 26 syllables from 20 birds taken (12 from *Zai et al., 2020b* and 8 additional ones) that were deafened and then recorded without any further manipulation. We used these birds to discount for pitch changes in WNd and dLO birds due to the absence of auditory feedback (see 'Bootstrapping').

### WN duration (WNdur)

Twelve birds underwent duration reinforcement using WN; in nine birds, the targeted sound feature was syllable duration, in two birds the targeted feature was syllable-plus-gap duration, and in one bird the targeted feature was gap duration. In four birds, the duration was squeezed, and in eight birds the duration was stretched. As in WNC birds, we did no further experimental manipulation after withdrawal of the WN stimulus. One bird changed its duration toward WN and showed an apparent appetitive response to WN as for the one muted bird.

### Muting

We muted birds by inserting a bypass cannula into the abdominal air sac (*Nilson et al., 2005*) as follows.

### Preparation of by-pass cannula

After incubation in 70% ethanol, we clogged a 7-mm-long polyimide tube (diameter 1.2 mm) with sterile paper tissue. We created a suture loop around the cannula and fixed the thread to the cannula with a knot and a drop of tissue glue.

### Cannula implantation

We anesthetized the birds with isoflurane (1.5–2%) and gave a single injection of carprofen (4 mg/kg). Subsequently, we applied local analgesic to the skin (2% lidocaine) and removed the feathers covering the right abdomen. We applied betadine solution on the exposed skin and made a small incision using sterilized scissors. We exposed the right abdominal air sac by shifting aside the fat tissue and

punctured it to create an opening. Immediately, we closed the opening by inserting the cannula and sealing the contact region with tissue glue. With the free end of the glued thread, we made one suture to the lowest rib. We closed the wound in the skin around the cannula with tissue glue and sutures using a new thread. Finally, we applied betadine solution on the wound and lidocaine gel around the injured site. Before releasing the bird to its cage, we removed the clog of the cannula with forceps and verified the air flow through the cannula.

We returned the birds to their home cage and monitored them for signs of suffering. We administered pain killers (meloxicam 2 mg/kg or carprofen 2–4 mg/kg) for 2 days after the surgery.

On the following days, we monitored the birds continuously for singing activity. If song was detected, the cannula was inspected for clogging and cleaned. Five birds unmuted spontaneously; they produced at most 300 songs before the bypass cannula was inspected and the clog was removed to re-mute the bird. To unclog the bypass cannulas, we used sharp forceps and sterile tissue dipped in saline. Six of eight birds produced quiet call-like vocalizations even on muted days on which no singing was detected.

## Deafening

We bilaterally removed cochleas as described in *Zai et al., 2020b*.

## Pitch analysis

In individual birds, we studied the dynamics of pitch recovery during the test period. In WNm birds, the test period started with unmuting, and in all other reinforced birds it started with the end of reinforcement. We analyzed songs in early ($E$) time windows defined as the first 2 hr window during the test period in which the bird produced at least 20 song motifs. We also assessed pitch recovery in late ($L$) windows defined exactly 4 days after the $E$ window. To make the measurements robust to circadian fluctuations of pitch, we compared the pitch values in early and late windows to pitch values produced in time-aligned windows during the last day of reinforcement ($R$) and during the last day of baseline ($B$).

We used this time-of-day matched analysis to produce *Figure 1H and I*, *Figure 2C and D*, and *Figure 3C and D*. Exceptions where time alignment was not possible are listed in the following:

- One WNm bird started singing late on the last day of reinforcement (preventing us from time-aligning the $R$ window with the $d'$ window), and therefore in this bird we defined $R$ after the end of WN but before muting (in this bird, there is more than 1 day of song after WN and before muting).
- In two birds (one WNC and one dLO bird), we defined the $L$ window one day earlier (on the fourth day, after 3 days of practice) because there was no data for these birds on the fifth day after reinforcement (our findings did not qualitatively change when we defined the $L$ window on the sixth day instead of the fourth).

- One WNm bird was housed together with a female during WN reinforcement; this bird did not sing during the time-match 2 hr period on the second, third, and fourth day after reinforcement; therefore, on those days, we computed the mean pitch from all values produced on that day in *Figure 1H*.

In early ($E$) and late ($L$) analysis windows, we computed the NRP, which is the remaining fraction of pitch shift since release from WN, defined as $\text{NRP}(X) = (P_X - P_B)/(P_R - P_B)$, where $P_X$ is either the mean pitch in the early ($X = E$) or late ($X = L$) window (*Figures 1H and I, 2C and D, and 3C and D*). $P_R$ and $P_B$ are the mean pitches in the $R$ and $B$ windows, respectively. An NRP of 33% indicates that two-thirds of the reinforced pitch shift have been recovered and an NRP of 0% indicates full recovery of baseline pitch. Note that the NRP measure discounts for differences in the amount of initial pitch shift the birds displayed at the beginning of the test period.

We performed statistical testing of NRP to discount for this diversity in initial pitch. To test the hypothesis that WNm birds recovered their baseline pitch without practice or that WNd or dLO birds recovered baseline pitch without auditory feedback, we performed a two-tailed $t$-test for NRP = 0.

Our results were qualitatively unchanged when we changed the timing of the $L$ window as long as there were at least 3 days between $E$ and $L$ windows (because WNC birds need at least 3 days to recover their baseline pitch in the $L$ window, p<0.05). Thus, giving deaf birds more time did not allow

them to recover their baseline pitch. Furthermore, we also tested larger windows of 4 and 24 hr duration instead of 2 hr and found qualitatively similar results. We further verified that our results did not critically depend on the time-alignment by repeating the NRP tests using the last 2 hr of reinforcement as the $R$ windows. Indeed, we found that all results in *Figures 1–3* were unchanged.

We computed the pitch change after reinforcement (*Figure 4*) as the difference in mean pitches between early ($E$) or late ($L$) and the last 2 hr of WN/LO reinforcement $R$ in units of sensitivity $d' = (P_X - P_R)/S_R$, where $S_R$ is the standard deviation of pitch values in the $R$ window. To test the hypothesis that WNd and dLO birds are able to make targeted pitch changes toward baseline, we performed a one-tailed $t$-test of the hypothesis H0: $d' < 0$. We used sensitivity $d'$ relative to the last 2 hr of WN/LO instead of NRP because we want to detect a pitch change, which is the realm of detection theory, that is, $d'$. Furthermore, by measuring local changes in pitch relative to the last 2 hr of WN/LO reinforcement, our measurements are only minimally affected by the amount of reinforcement learning that might have occurred during this 2 hr time window – choosing an earlier or longer window would have blended reinforced pitch changes into our estimates. Last but not least, changes in the way in which we normalized $d'$ values – dividing by $S_B$, $\sqrt{(S_B^2 + S_R^2)/2}$, or $\sqrt{(S_X^2 + S_R^2)/2}$ – or using the NRP relative to the last 2 hr of WN/LO did not qualitatively change the results shown in *Figure 4D*.

## Test for recovery of baseline after duration reinforcement and pitch reinforcement

Zebra finches learn both syllable pitch and syllable duration from their tutors (*Lipkind and Tchernichovski, 2011*; *Glaze and Troyer, 2013*). We tested whether these distinct sound features are equally well maintained in adulthood. Both pitch and duration can be driven away from baseline using aversive reinforcers (*Ali et al., 2013*). To test whether recovery of syllable duration is similarly fast as recovery of pitch (*Canopoli et al., 2016*), we induced birds to shorten or lengthen a targeted song syllable by playing a loud WN stimulus when the syllable duration was shorter or longer than a manually set threshold. However, we found that unlike for pitch, after withdrawal of reinforcement, these WNdur birds did not reliably recover their baseline syllable duration: after 4 days of song practice without reinforcement, their average normalized residual syllable duration (NRD) – the remaining fraction of duration difference to baseline since release – was significantly different from zero (NRD = 42%, p=0.007, two-sided $t$-test for NRD = 0, N = 12 birds, *Figure 1—figure supplement 1*), suggesting that adult birds do not maintain syllable duration as rigidly as they do maintain pitch.

Indeed, in contrast, pitch-reinforced (WNC) birds fully recovered baseline pitch 4 days after withdrawal of reinforcement (average NRP = 1%, p=0.91 two-sided $t$-test for NRP = 0, N = 18 birds, *Figure 1—figure supplement 1B and C*, same data as in *Figure 1H and I*). Moreover, the recovery of pitch was more extensive than that of duration: after 4 days, the average NRP was smaller than the average NRD (p=0.001, tstat = -3.56, df = 28, two-tailed $t$-test). The difference between NRP and NRD persisted over time and was still present after 10 days of practice (NRP = -2% and NRD = 33% after 10 days of practice, p=0.04, tstat = -2.26, df = 18, N = 10 WNC and N = 10 WNdur birds recorded 11 days after WN, two-tailed $t$-test), showing that syllable duration is not recovered as reliably as pitch after withdrawal of WN reinforcement.

WNC birds received WN reinforcement during 11 ± 3 days on average (range 8–15 days) and during this period they changed their pitch by 0.24 ± 0.22 $d'$/day (total shift 2.5 ± 1.8 $d'$) on average, which did not significantly differ from the behavior of WNdur birds (p = 0.44, tstat = 0.78, df = 28, two-tailed $t$-test) that received WN reinforcement during 10 ± 5 days on average (range 5–23 days) and changed syllable duration during this period by 0.36 ± 0.17 $d'$/day (total shift 2.9 ± 0.7 $d'$) on average (p = 0.14, tstat = -1.49, df = 28, two-tailed $t$-test). Furthermore, we did not find a correlation between the number of days with reinforcement and the NRP/NRD after 4 days of practice in either WNC birds (correlation coefficient R = 0.20, p = 0.43), WNdur birds (R = 0.05, p=0.88), or in both groups combined (R = -0.01, p = 0.94), nor between the total shift and the NRP/NRD after 4 days of practice in either WNC birds (R = 0.34, p = 0.17), WNdur birds (R = -0.12, p=0.71), or in both groups combined (R = 0.18, p = 0.34), suggesting that the difference in recovery behavior between WNdur and WNC birds is neither due to differences in time spent with reinforcement nor to differences in learning speed, but due to differences in attachment to these sound features.

This difference in recovery behavior is consistent with the literature. *Ali et al., 2013* found that pitch recovers at roughly half the rate it changed during reinforcement (12.1 Hz/day during recovery

vs. 26.0 Hz/day during reinforcement). By contrast, they found that duration (of mostly syllables + gaps) reverted toward baseline at only about a quarter of the rate measured during reinforcement (0.8 ms/day vs. 3.4 ms/day) (*Ali et al., 2013*), suggesting that recovery of duration proceeds at a much slower pace than birds are capable of. Similarly, *Roberts et al., 2017* found pitch recovered faster than it changed during reinforcement, whereas syllable + gap durations recovered slower than they changed during reinforcement five birds each. Both studies used only a few birds and did not directly compare the recovery of duration versus pitch. Also, the published reports preclude a comparison in terms of the unit-free *d'* measure post hoc because the baseline pitch and the duration variability are not reported in these studies.

Interestingly, in the context of *Figure 4*, duration reverts similarly to pitch. After 4 days of practice WNdur birds showed a significant reversion toward baseline ($d'$ = -2.03, p = 5.5 · 10$^{-4}$, tstat = -4.37, N = 11 WNdur birds, one-sided *t*-test), indicating that WNdur birds are capable of revertive changes beyond the planning limit ($d' \simeq 1$) found for pitch reversion. Thus, it remains unclear whether the difference in NRP and NRD recovery (*Figure 1—figure supplement 1*) could be confounded by the disparity in experimental parameters (i.e., difference in total shifts). Nevertheless, all studies (*Ali et al., 2013*; *Roberts et al., 2017*) and our analysis concur that the recovery of baseline pitch is more efficient than the recovery of duration, suggesting that duration is less accurately maintained than pitch. We therefore decided to perform all experiments using pitch as the targeted song feature and not syllable duration.

## Bootstrapping

To test whether deaf birds indeed make small pitch changes toward a target if and only if they experienced target-mismatch during reinforcement, we bootstrapped the difference in pitch changes between reinforced (WNd and dLO) and deaf control birds (dC). All dC birds were recorded for at least 5 days after they started singing while deaf.

In dC birds, we defined the $R$, $E$, and $L$ windows such that they matched those of WNd and dLO birds in terms of days since deafening. Additionally, in dLO birds we chose the windows such that they matched in terms of time of day (because LO always ended overnight). Thus, the $R$ windows in dC birds either corresponded to the last 2 hr before deafening (as control for WNd birds) or to the last 2 hr of the day before $E$ (as control for dLO birds).

For WNd birds, we obtained in total 26 control syllables from 20 dC birds. For dLO birds, we obtain 17 control syllables from 13 dC birds (some dC birds did not provide any useable data because they stopped singing or were not recorded for long enough).

For the bootstrapping procedure, we randomly paired control syllables (N = 26 for WNd and N = 17 for dLO) one-by-one with matchable syllables from reinforced birds (with replacement), computed the mean pitches $P_R$, $P_E$, $P_L$ in corresponding windows, calculated the standard deviation $S_R$, calculated the average pitch changes $d'_E = (P_E - P_R)/S_R$ and $d'_L = (P_L - P_R)/S_R$ for both manipulated and control birds, and multiplied these by –1 if the reinforced bird was down-shifted (as we did for *d'* above). We then took the differences in average pitch changes between manipulated (WNd and dLO) and dC birds, for example, $d'_{E,WNd} - d'_{E,dC}$. We repeated this procedure 10,000 times and plotted the distribution of average pitch change differences between WNd and dC (red) and between dLO and dC (blue) in *Figure 4E* and perform bootstrap statistics.

Our results were qualitatively unchanged (only WNd significantly reverted pitch toward baseline) when we aligned the $R$ windows by the time of day of the corresponding $E$ windows (two dC birds started singing later on the day of the $E$ window than they stopped singing on the days before; in these two birds, we used the $R$ windows instead; see *Supplementary file 1*). Although the *d'* values in both groups increased (and in dLO birds, the average *d'* in the $L$ windows was positive, p<0.05, two-tailed *t*-test), we found a significant pitch difference between WNd or dLO birds in $L$ windows, which upholds our findings that mismatch experience is necessary for pitch reversion. The reason for the increases in *d'* likely is that birds further shifted their pitch away from baseline on the last day of reinforcement (after the time-aligned $R$ window). Also, results were robust when we analyzed pitch changes after release from reinforcement in units of NRP: without practice, WNd birds made small and significant pitch changes toward baseline, and dLO birds stayed at NRP ≥ 1.

## Linear mixed effect model

We simulated a linear mixed effect model on the combined NRP data from all groups with fixed effects corresponding to time (general offset $a$, late $b$), treatment (deafened $c$, muted $d$), mismatch-experience

$e$, and a fixed effect $f$ that is linear in the time between the time $t(R)$ of the $R$ window and the time $t(E)$ of the $E$ window. We used this latter term to test whether birds can recover without practice. We further included a fixed effect in terms of the interaction $g$ between deafening and late, to test whether birds recover without auditory feedback (but with practice from $E$ to $L$). The equation for the NRP $N_{i,t}$ in bird $i$ in time window $t$

$$N_{i,t} = a + b\delta_{t \in L} + c\delta_{i \in WNd \text{ or } i \in dLO} + d\delta_{i \in WNm} + e\delta_{i \notin dLO} + f(t(E) - t(R))\delta_{t \in E} + g\delta_{i \in WNd \text{ or } i \in dLO}\delta_{i \in L} + \varepsilon_{i l \text{group}}.$$

The Kronecker $\delta$ of a specific group equals 1 if bird $i$ belongs to that group (e.g., $i \in WN$) resp. if the time window $t$ is either $E$ or (e.g. $t \in L$), and it equals 0 otherwise. The terms $\varepsilon_{i l \text{group}}$ is a random effect associated with a particular bird $i$ and group. Note that the fixed effect of mismatch experience $e$ is 0 for dLO birds and 1 for all other birds.

## Acknowledgements

We thank Manon Rolland and Sophie Cavé-Lopez for performing some of the deafening surgeries and their support with the experiments, and Heiko Hörster for providing excellent animal breeding and animal care services.

## Additional information

### Funding

| Funder | Grant reference number | Author |
|---|---|---|
| Swiss National Science Foundation | 31003A_182638 | Anja T Zai<br>Anna E Stepien<br>Richard HR Hahnloser |
| Swiss National Science Foundation | 31003A_156976/1 | Anja T Zai<br>Anna E Stepien<br>Richard HR Hahnloser |
| European Research Council | 268911 VOTECOM | Anja T Zai<br>Anna E Stepien<br>Richard HR Hahnloser |

The funders had no role in study design, data collection and interpretation, or the decision to submit the work for publication.

### Author contributions

Anja T Zai, Conceptualization, Data curation, Software, Formal analysis, Investigation, Visualization, Methodology, Writing - original draft, Writing - review and editing; Anna E Stepien, Conceptualization, Data curation, Investigation, Visualization, Methodology, Writing - original draft, Writing - review and editing; Nicolas Giret, Supervision, Funding acquisition, Investigation, Writing - review and editing; Richard HR Hahnloser, Conceptualization, Formal analysis, Supervision, Funding acquisition, Visualization, Writing - original draft, Writing - review and editing

### Author ORCIDs

Anja T Zai 
Anna E Stepien 
Nicolas Giret 
Richard HR Hahnloser 

### Ethics

All experimental procedures were in accordance with the Veterinary Office of the Canton of Zurich (licenses 123/2010 and 207/2013) or by the French Ministry of Research and the ethical committee Paris-Sud and Centre (CEEA N°59, project 2017-12).

Reviewer #1 (Public Review): https://doi.org/10.7554/eLife.90445.4.sa1
Reviewer #3 (Public Review): https://doi.org/10.7554/eLife.90445.4.sa2
Author response https://doi.org/10.7554/eLife.90445.4.sa3

## Additional files

### Supplementary files
MDAR checklist

Supplementary file 1. Bootstrapped pitch differences between reinforced and deaf control birds.

### Data availability
Pitch and duration data that support the findings of this study together with the MATLAB scripts to reproduce the analysis and figures are available at the ETH Research Collection at https://doi.org/10.3929/ethz-b-000670443. The raw audio data underlying the pitch measurement is deposited on Zenodo at https://doi.org/10.5281/zenodo.14732250.

The following datasets were generated:

| Author(s) | Year | Dataset title | Dataset URL | Database and Identifier |
| --- | --- | --- | --- | --- |
| Zai AT, Stepien AE, Giret N, Hahnloser RHR | 2024 | Goal-directed vocal planning in a songbird - dataset | https://doi.org/10.3929/ethz-b-000670443 | ETH Library research collection, 10.3929/ethz-b-000670443 |
| Zai AT, Stepien AE, Giret N, Hahnloser RHR | 2025 | Goal-directed vocal planning in a songbird - raw data | https://doi.org/10.5281/zenodo.14732250 | Zenodo, 10.5281/zenodo.14732250 |

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
