## [Editor Report · eLife assessment]

This **important** work identifies a previously uncharacterized capacity for songbird to recover vocal targets even without sensory experience. The evidence supporting this claim is **convincing**, with technically difficult and innovative experiments exploring goal-directed vocal plasticity in deafened birds. This work has broad relevance to the fields of vocal and motor learning.

---

## [Referee Report · Reviewer #1 (Public Review)]

Summary:

Zai et al test if songbirds can recover the capacity to sing auditory targets without singing experience or sensory feedback. Past work showed that after the pitch of targeted song syllables are driven outside of birds' preferred target range with external reinforcement, birds revert to baseline (i.e. restore their song to their target). Here the authors tested the extent to which this restoration occurs in muted or deafened birds. If these birds can restore, this would suggest an internal model that allows for sensory-to-motor mapping. If they cannot, this would suggest that learning relies entirely on feedback dependent mechanisms, e.g. reinforcement learning (RL). The authors find that deafened birds exhibit moderate but significant restoration, consistent with the existence of a previously under-appreciated internal model in songbirds.

Strengths:

The experimental approach of studying vocal plasticity in deafened or muted birds is innovative, technically difficult and perfectly suited for the question of feedback-independent learning. The finding in Figure 4 that deafened birds exhibit subtle but significant plasticity toward restoration of their pre-deafening target is surprising and important for the songbird and vocal learning fields, in general.

In this revision, the authors suitably addressed confusion about some statistical methods related to Fig. 4, where the main finding of vocal plasticity in deafened birds was presented.

There remain minor issues in the presentation early in the results section and in Fig. 4 that should be straightforward to clarify in the revision.

---

## [Referee Report · Reviewer #3 (Public Review)]

Summary:

Zai et al. test whether birds can modify their vocal behavior in a manner consistent with planning. They point out that while some animals are known to be capable of volitional control of vocalizations, it has been unclear if animals are capable of planning vocalizations-that is, modifying vocalizations towards a desired target without the need to learn this modification by practising and comparing sensory feedback of practised behavior to the behavioral target. They study zebra finches that have been trained to shift the pitch of song syllables away from their baseline values. It is known that once this training ends, zebra finches have a drive to modify pitch so that it is restored back to its baseline value. They take advantage of this drive to ask whether birds can implement this targeted pitch modification in a manner that looks like planning, by comparing the time course and magnitude of pitch modification in separate groups of birds who have undergone different manipulations of sensory and motor capabilities. A key finding is that birds who are deafened immediately before the onset of this pitch restoration paradigm, but after they have been shifted away from baseline, are able to shift pitch partially back towards their baseline target. In other words, this targeted pitch shift occurs even when birds don't have access to auditory feedback, which argues that this shift is not due to reinforcement-learning-guided practice, but is instead planned based on the difference between an internal representation of the target (baseline pitch) and current behavior (pitch the bird was singing immediately before deafening).

The authors present additional behavioral studies arguing that this pitch shift requires auditory experience of song in its state after it has been shifted away from baseline (birds deafened early on, before the initial pitch shift away from baseline, do not exhibit any shift back towards baseline), and that a full shift back to baseline requires auditory feedback. The authors synthesize these results to argue that different mechanisms operate for small shifts (planning, which does not need auditory feedback) and large shifts (through a mechanism that requires auditory feedback).

The authors also make a distinction between two kinds of planning: covert-not requiring any motor practice and overt-requiring motor practice but without access to auditory experience from which target mismatch could be computed. They argue that birds plan overtly, based on these deafening experiments as well as an analogous experiment involving temporary muting, which suggests that indeed motor practice is required for pitch shifts.

Strengths:

The primary finding (that partially restorative pitch shift occurs even after deafening) rests on strong behavioral evidence. It is less clear to what extent this shift requires practice, since their analysis of pitch after deafening takes the average over within the first two hours of singing. If this shift is already evident in the first few renditions then this would be evidence for covert planning. Technical hurdles, such as limited sample sizes and unstable song after surgical deafening, make this difficult to test. (Similarly, the authors could test whether the first few renditions after recovery from muting already exhibit a shift back towards baseline.)

This work will be a valuable addition to others studying birdsong learning and its neural mechanisms. It documents features of birdsong plasticity that are unexpected in standard models of birdsong learning based on reinforcement and are consistent with an additional, perhaps more cognitive, mechanism involving planning. As the authors point out, perhaps this framework offers a reinterpretation of the neural mechanisms underlying a prior finding of covert pitch learning in songbirds (Charlesworth et al., 2012).

A strength of this work is the variety and detail in its behavioral studies, combined with sensory and motor manipulations, which on their own form a rich set of observations that are useful behavioral constraints on future studies.

Weaknesses:

The argument that pitch modification in deafened birds requires some experience hearing their song in its shifted state prior to deafening (Fig. 4) is solid but has an important caveat. Their argument rests on comparing two experimental conditions: one with and one without auditory experience of shifted pitch. However, these conditions also differ in the pitch training paradigm: the "with experience" condition was performed using white noise training, while the "without experience" condition used "lights off" training (Fig. 4A). It is possible that the differences in ability for these two groups to restore pitch to baseline reflects the training paradigm, not whether subjects had auditory experience of the pitch shift. Ideally, a control study would use one of the training paradigms for both conditions, which would be "lights off" or electrical stimulation (McGregor et al. 2022), since WN training cannot be performed in deafened birds. In the Discussion, in response to this point, the authors point out that birds are known to recover their pitch shift if those shifts are driven using electrical stimulation as reinforcement (McGregor et al. 2022); however, it is arguably still relevant to know whether a similar recovery occurs for the "lights off" paradigm used here.

---

## [Author Response]

The following is the authors’ response to the previous reviews

**Reviewer #1 (Recommendations For The Authors):**
In this revision the authors address some of the key concerns, including clarification of the balanced nature of the RL driven pitch changes and conducting analyses to control for the possible effects of singing quantity on their results. The paper is much improved but still has some sources of confusion, especially around Fig. 4, that should be fixed. The authors also start the paper with a statistically underpowered minor claim that seems unnecessary in the context of the major finding. I recommend the authors may want to restructure their results section to focus on the major points backed by sufficient n and stats.Major issues.(1) The results section begins very weak - a negative result based on n=2 birds and then a technical mistake of tube clogging re-spun as an opportunity to peak at intermittent song in the otherwise muted birds. The logic may be sound but these issues detract from the main experiment, result, analysis, and interpretation. I recommend re-writing this section to home in on, from the outset, the well-powered results. How much is really gained from the n=2 birds that were muted before ANY experience? These negative results may not provide enough data to make a claim. Nor is this claim necessary to motivate what was done in the next 6 birds. I recommend dropping the claim?

We thank the reviewer for the recommendation. We moved the information to the Methods.

(2) Fig. 4 is very important yet remains very confusing, as detailed below.Fig. 4a. Can the authors clarify if the cohort of WNd birds that give rise to the positive result in Fig 4 ever experienced the mismatch in the absence of ongoing DAF reinforcement pre-deafening? Fig4a does nor the next clearly specifies this. This is important because we know that there are day timescale delays in LMAN-dependent bias away from DAF and consolidation into the HVC-RA pathway (Andalman and Fee, 2009). Thus, if birds experienced mismatch pre-deafening in the absence of DAF, then an earnly learning phase in Area X could be set in place. Then deafening occurs, but these weight changes in X could result in LMAN bias that expresses only days later -independent of auditory feedback. Such a process would not require an internal model as the authors are arguing for here. It would simply arise from delays in implementing reinforcement-driven feedback. If the birds in Fig 4 always had DAF on before deafening, then this is not an issue. But if the birds had hours of singing with DAF off before deafening, and therefore had the opportunity to associate DA error signals with the targeted time in the song e.g. pauses on the far-from-target renditions (Duffy et al, 2022), then the return-to-baseline would be expected to be set in place independent of auditory feedback. Please clarify exactly if the pitch-contingent DAF was on or off in the WNd cohort in the hours before deafening. In Fig. 3b it looks like the answer is yes but I cannot find this clearly stated in the text.

We did not provide DAF-free singing experience to the birds in Fig. 4 before deafening. Thus, according to the reviewer, the concern does not apply.

Note that we disagree with the reviewer’s premise that there is ‘day timescale delay in LMAN-dependent bias away from DAF and consolidation into the HVC-RA pathway’. More recent data reveals immediate consolidation of the anterior forebrain bias without a night-time effect (Kollmorgen, Hahnloser, Mante 2020; Tachibana, Lee, Kai, Kojima 2022). Thus, the single bird in (Andalman and Fee 2009) seems to be somewhat of an outlier.

Hearing birds can experience the mismatch regardless of whether they experience DAF-free singing (provided their song was sufficiently shifted): even the renditions followed by white noise can be assessed with regards to their pitch mismatch, so that DAF imposes no limitation on mismatch assessment.

We disagree with their claim that no internal model would be needed in case consolidation was delayed in Area X. If indeed, Area X stores the needed change and it takes time to implement this change in LMAN, then we would interpret the change in Area X as the plan that birds would be able to implement without auditory feedback. Because pitch can either revert (after DAF stops) or shift further away (when DAF is still present), there is no rigid delay that is involved in recovering the target, but a flexible decision making of implementing the plan, which in our view amounts to using a model.

Fig 4b. Early and Late colored dots in legend are both red; late should be yellow? Perhaps use colors that are more distinct - this may be an issue of my screen but the two colors are difficult to discern.

We used colors yellow to red to distinguish different birds and not early and late. We modified the markers to improve visual clarity: Early is indicated with round markers and late with crosses.

Fig 4b. R, E, and L phases are only plotted for 4c; not in 4b. But the figure legend says that R, E and L are on both panels.

In Fig. 4b E and L are marked with markers because they are different for different birds. In Fig. 4c the phases are the same for all birds and thus we labeled them on top. We additionally marked R in Fig. 4b as in Fig. 4c.

Fig 4e. Did the color code switch? In the rest of Fig 4, DLO is red and WND is blue. Then in 4e it swaps. Is this a typo in the caption? Or are the colors switch? Please fix this it's very confusing.

Thank you for pointing out the typo in the caption. We corrected it.

The y axes in Fig 4d-e are both in std of pitch change - yet they have different ylim which make it visually difficult to compare by eye. Is there a reason for this? Can the authors make the ylim the same for fig 4d-e?.

We added dashed lines to clarify the difference in ylim.

Fig 4d-3 is really the main positive finding of the paper. Can the others show an example bird that showcases this positive result, plotted as in Fig 3b? This will help the audience clearly visualize the raw data that go into the d' analyses and get a more intuitive sense of the magnitude of the positive result.

We added example birds to figure 4, one for WNd and one for dLO.

Please define 'late' in Fig.4 legend.

Done

MinorDefine NRP In the text with an example. Is an NRP of 100 where the birds was before the withdrawal of reinforcement?

We added the sentence to the results:

"We quantified recovery in terms of 𝑵𝑹𝑷 to discount for differences in the amount of initial pitch shift where 𝑵𝑹𝑷 = 𝟎% corresponds to complete recovery and 𝑵𝑹𝑷 = 𝟏𝟎𝟎% corresponds pitch values before withdrawal of reinforcement (R) and thus no recovery."

**Reviewer #3 (Recommendations For The Authors):**
The use of "hierarchically lower" to refer to the flexible process is confusing to me, and possibly to many readers. Some people think of flexible, top-down processes as being _higher_ in a hierarchy. Regardless, it doesn't seem important, in this paper, to label the processes in a hierarchy, so perhaps avoid using that terminology.

We reformulated the paragraph using ‘nested processes’ instead of hierarchical processes.

In the statement "a seeming analogous task to re-pitching of zebra finch song, in humans, is to modify developmentally learned speech patterns", a few suggestions: it is not clear whether "re-pitching" refers to planning or feedback-dependent learning (I didn't see it introduced anywhere else). And if this means planning, then it is not clear why this would be analogous to "humans modifying developmentally learned speech patterns". As you mentioned, humans are more flexible at planning, so it seems re-pitching would _not_ be analogous (or is this referring to the less flexible modification of accents?).

We changed the sentence to:

"Thus, a seeming analogous task to feedback-dependent learning of zebra finch song, in humans, is to modify developmentally learned speech patterns."